# Kernelized Heterogeneous Risk Minimization

**Jiashuo Liu**[1,*], **Zheyuan Hu**[2,*], **Peng Cui**[1,†], **Bo Li**[3] **and Zheyan Shen**[1]
[1] Department of Computer Science & Technology, Tsinghua University, Beijing, China
[2]Department of Computer Science, National University of Singapore, Singapore
[3]School of Economics and Management, Tsinghua University, Beijing, China
{liujiashuo77,zyhu2001}@gmail.com, cuip@tsinghua.edu.cn,
libo@sem.tsinghua.edu.cn, shenzy17@mails.tsinghua.edu.cn

## Abstract

The ability to generalize under distributional shifts is essential to reliable machine learning, while models optimized with empirical risk minimization usually fail on non-$i.i.d$ testing data. Recently, invariant learning methods for out-of-distribution (OOD) generalization propose to find causally invariant relationships with multi-environments. However, modern datasets are frequently multi-sourced without explicit source labels, rendering many invariant learning methods inapplicable. In this paper, we propose Kernelized Heterogeneous Risk Minimization (KerHRM) algorithm, which achieves both the latent heterogeneity exploration and invariant learning in kernel space, and then gives feedback to the original neural network by appointing invariant gradient direction. We theoretically justify our algorithm and empirically validate the effectiveness of our algorithm with extensive experiments.

## 1 Introduction

Traditional machine learning algorithms which optimize the empirical risk often suffer from poor generalization performance under distributional shifts caused by latent heterogeneity or selection biases that widely exist in real-world data[9, 20]. How to guarantee a machine learning algorithm with good generalization ability on data drawn out-of-distribution is of paramount significance, especially in high-stake applications such as financial analysis, criminal justice and medical diagnosis, etc.[13, 18], which is known as the out-of-distribution(OOD) generalization problem[1].

To ensure the OOD generalization ability, invariant learning methods assume the existence of the causally invariant correlations and exploit them through given environments, which makes their performances heavily dependent on the quality of environments. Further, the requirements for the environment labels are too strict to meet with, since real-world datasets are frequently assembled by merging data from multiple sources without explicit source labels. Recently, several works[5, 15] to relax such restrictions have been proposed. Creager *et al.*[5] directly infer the environments according to a given biased model first and then performs invariant learning. But the two stages cannot be jointly optimized and the quality of inferred environments depends heavily on the pre-provided biased model. Further, for complicated data, using invariant representation for environment inference is harmful, since the environment-specific features are gradually discarded, causing the extinction of latent heterogeneity and rendering data from different latent environments undistinguishable. Liu *et al.*[15] design a mechanism where two interactive modules for environment inference and invariant learning respectively can promote each other. However, it can only deal with scenarios where invariant and variant features are decomposed on raw feature level, and will break down when the decomposition can only be performed in representation space(e.g., image data).

---

*Equal Contributions
†Corresponding Author

35th Conference on Neural Information Processing Systems (NeurIPS 2021).

This paper focuses on the integration of latent heterogeneity exploration and invariant learning on representation level. In order to incorporate representation learning with theoretical guarantees, we introduce Neural Tangent Kernel(NTK[10]) into our algorithm. According to NTK theory[10], training the neural network is equivalent to linear regression using Neural Tangent Features(NTF), which converts non-linear neural networks into linear regression in NTF space and makes the integration possible. Based on this, our Kernelized Heterogeneous Risk Minimization (KerHRM) algorithm is proposed, which synchronously optimizes the latent heterogeneity exploration module $\mathcal{M}_c$ and invariance learning module $\mathcal{M}_p$ in NTF space. Specifically, we propose our novel Invariant Gradient Descent(IGD) for $\mathcal{M}_p$, which performs invariant learning in NTF space and then feeds back to neural networks with appointed invariant gradient direction. For $\mathcal{M}_c$, we construct an orthogonal heterogeneity-aware kernel to capture the environment-specific features and to further accelerate the heterogeneity exploration. Theoretically, we demonstrate our heterogeneity exploration algorithm for $\mathcal{M}_c$ with rate-distortion theory and justify the orthogonality property of the built kernel, which jointly can illustrate the mutual promotion between the two modules. Empirically, experiments on both synthetic and real-world data validate the superiority of KerHRM in terms of good out-of-distribution generalization performance.

## 2 Preliminaries

Following [1, 3], we consider data $D = \{D^e\}_{e \in \mathrm{supp}(\mathcal{E}_{tr})}$ with different sources data $D^e = \{X^e, Y^e\}$ collected from multiple training environments $\mathcal{E}_{tr}$. Here environment labels are unavailable as in most of the real applications. $\mathcal{E}_{tr}$ is a random variable on indices of training environments and $P^e$ is the distribution of data and label in environment $e$. The goal of this work is to find a predictor $f(\cdot) : \mathcal{X} \to \mathcal{Y}$ with good out-of-distribution generalization performance, which is formalized as:

$$\arg\min_f \max_{e \in \mathrm{supp}(\mathcal{E})} \mathcal{L}(f|e) \tag{1}$$

where $\mathcal{L}(f|e) = \mathbb{E}^e[\ell(X^e, Y^e)]$ represents the risk of predictor $f$ on environment $e$, and $\ell(\cdot, \cdot) : \mathcal{Y} \times \mathcal{Y} \to \mathbb{R}^+$ the loss function. Note that $\mathcal{E}$ is the random variable on indices of all possible environments such that $\mathrm{supp}(\mathcal{E}_{tr}) \subset \mathrm{supp}(\mathcal{E})$. Usually, for all $e \in \mathrm{supp}(\mathcal{E}) \setminus \mathrm{supp}(\mathcal{E}_{tr})$, the data and label distribution $P^e(X, Y)$ can be quite different from that of training environments $\mathcal{E}_{tr}$. Therefore, the problem in equation 1 is referred to as Out-of-Distribution (OOD) Generalization problem [1]. Since it is impossible to characterize the latent environments $\mathcal{E}$ without any prior knowledge or structural assumptions, the invariance assumption is proposed for invariant learning:

**Assumption 2.1.** *There exists random variable $\Psi_S^*(X)$ such that the following properties hold:*
*a.* Invariance property*: for all $e, e' \in \mathrm{supp}(\mathcal{E})$, we have $P^e(Y|\Psi_S^*(X)) = P^{e'}(Y|\Psi_S^*(X))$ holds.*
*b.* Sufficiency property*: $Y = f(\Psi_S^*) + \epsilon, \ \epsilon \perp X$.*

This assumption indicates invariance and sufficiency for predicting the target $Y$ using $\Psi_S^*$, which is known as invariant representations with stable relationships with $Y$ across $\mathcal{E}$. To acquire such $\Psi_S^*$, a branch of works[4, 11, 15] proposes to find the maximal invariant predictor(MIP) of an invariance set, which are defined as follows:

**Definition 2.1.** *The **invariance set** $\mathcal{I}$ with respect to $\mathcal{E}$ is defined as:*

$$\mathcal{I}_{\mathcal{E}} = \{\Psi_S(X) : Y \perp \mathcal{E}|\Psi_S(X)\} = \{\Psi_S(X) : H[Y|\Psi_S(X)] = H[Y|\Psi_S(X), \mathcal{E}]\} \tag{2}$$

*where $H[\cdot]$ is the Shannon entropy of a random variable. The corresponding **maximal invariant predictor (MIP)** of $\mathcal{I}_{\mathcal{E}}$ is defined as $S = \arg\max_{\Phi \in \mathcal{I}_{\mathcal{E}}} \mathbb{I}(Y; \Phi)$, where $\mathbb{I}(\cdot; \cdot)$ measures Shannon mutual information between two random variables.*

Firstly, we propose that using the maximal invariant predictor $S$ of $\mathcal{I}_{\mathcal{E}}$ can guarantee OOD optimality in Theorem 2.1. The formal statement is similar to [15] and can be found in Appendix A.3.

**Theorem 2.1.** *(Optimality Guarantee, informal) For predictor $\Phi^*(X)$ satisfying Assumption 2.1, $\Psi_S^*$ is the maximal invariant predictor with respect to $\mathcal{E}$ and the solution to OOD problem in equation 1 is $\mathbb{E}_Y[Y|\Psi_S^*] = \arg\min_f \sup_{e \in \mathrm{supp}(\mathcal{E})} \mathbb{E}[\mathcal{L}(f)|e]$.*

However, recent works[4, 11] on finding MIP solutions rely on the availability of data from multiple training environments $\mathcal{E}_{tr}$, which is hard to meet with in practice. Further, their validity is highly determined by the given $\mathcal{E}_{tr}$. Since $\mathcal{I}_{\mathcal{E}} \subseteq \mathcal{I}_{\mathcal{E}_{tr}}$, the invariance regularized by $\mathcal{E}_{tr}$ is often too large and the learned MIP may contain variant components and fails to generalize well. Based on this,

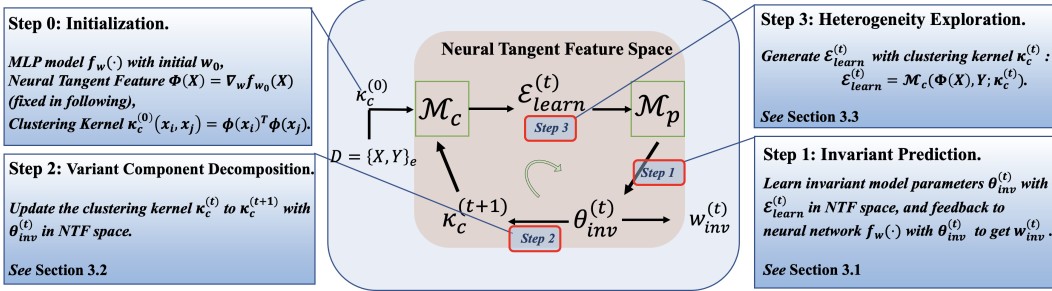

Figure 1: The framework for KerHRM. The middle block diagram shows the overall flow of the algorithm, which consists of two modules named heterogeneity exploration module $\mathcal{M}_c$ and invariant prediction module $\mathcal{M}_p$. The whole algorithm runs iteratively between $\mathcal{M}_c$ and $\mathcal{M}_p$, where one iteration consists of three steps, which we illustrate in section 3.1, 3.2 and 3.3 respectively.

Heterogeneous Risk Minimization(HRM[15]) proposes to generate environments $\mathcal{E}_{tr}$ with minimal $|\mathcal{I}_{\mathcal{E}_{tr}}|$ and to conduct invariant prediction with learned $\mathcal{E}_{tr}$. However, the proposed HRM can only deal with simple scenarios where $X = [\Psi_S^*, \Psi_V^*]^T$ on raw feature level ($\Psi_S^*$ are invariant features and $\Psi_V^*$ variant ones), and will break down where $X = h(\Psi_S^*, \Psi_V^*)$ ($h(\cdot, \cdot)$ is an unknown transformation function), since the decomposition can only be performed in representation space. In this work, we focus on the integration of latent heterogeneity exploration and invariant learning in general scenarios where invariant features are latent in $X$, which can be easily fulfilled in real applications.

**Problem 1.** *(Problem Setting)    Assume that $X = h(\Psi_S^*, \Psi_V^*) \in \mathbb{R}^d$, where $\Psi_S^*$ satisfies Assumption 2.1, $h(\cdot)$ is an unknown transformation function and $\Psi_S^* \perp \Psi_V^*$ (following functional representation lemma[7]), given heterogeneous dataset $D = \{D^e\}_{e \in \mathrm{supp}(\mathcal{E}_{latent})}$ without environment labels, **the task** is to generate environments $\mathcal{E}_{learn}$ with minimal $|\mathcal{I}_{\mathcal{E}_{learn}}|$ and meanwhile learn invariant models.*

## 3   Method

**Remark.** *Following the analysis in section 2, to generate environments $\mathcal{E}_{learn}$ with minimal $|\mathcal{I}_{\mathcal{E}_{learn}}|$ is equivalent to generate environments with as varying $P(Y|\Psi_V^*)$ as possible, so as to exclude variant parts $\Psi_V^*$ from the invariant set $\mathcal{I}_{\mathcal{E}_{learn}}$.*

In spite of such insight, the latent $\Psi_S^*, \Psi_V^*$ make it impossible to directly generate $\mathcal{E}_{learn}$. In this work, we propose our Kernelized Heterogeneous Risk Minimization (KerHRM) algorithm with two interactive modules, the frontend $\mathcal{M}_c$ for heterogeneity exploration and backend $\mathcal{M}_p$ for invariant prediction. Specifically, given pooled data, the algorithm starts with the heterogeneity exploration module $\mathcal{M}_c$ with a learned heterogeneity-aware kernel $\kappa_c$ to generate $\mathcal{E}_{learn}$. The learned environments are used by $\mathcal{M}_p$ to produce invariant direction $\theta_{inv}$ in Neural Tangent Feature(NTF) space that captures the invariant components $\Psi_S$, and then $\theta_{inv}$ is used to guide the gradient descent of neural networks. After that, we update the kernel $\kappa_c$ to orthogonalize with the invariant direction $\theta_{inv}$ so as to better capture the variant components $\Psi_V$ and realize the mutual promotion between $\mathcal{M}_c$ and $\mathcal{M}_p$ iteratively. The whole framework is jointly optimized, so that the mutual promotion between heterogeneity exploration and invariant learning can be fully leveraged. For smoothness we

---

**Algorithm 1** Kernelized Heterogeneous Risk Minimization (KerHRM) Algorithm

**Input:** Heterogeneous dataset $D = \{D^e = (X^e, Y^e)\}_{e \in \mathcal{E}_{tr}}$
**Initialization:** MLP model $f_w(\cdot)$ with initialized $w_0$, Neural Tangent Feature $\Phi(X) = \nabla_w f_{w_0}(X)$(fixed in following), clustering kernel initialized as $\kappa_c^{(0)}(x_1, x_2) = x_1^T x_2$
**for** $t = 1$ **to** $T$ **do**
    1. Generate $\mathcal{E}_{learn}^{(t)}$ with clustering kernel $\kappa_c^{(t-1)}$: $\mathcal{E}_{learn}^{(t)} = \mathcal{M}_c((\Phi(X), Y), \kappa_c^{(t-1)})$
    2. Learn invariant model parameters $\theta_{inv}^{(t)}$ with $\mathcal{E}_{learn}^{(t)}$ in NTF space: $\theta_{inv}^{(t)} = \mathcal{M}_p(\mathcal{E}_{learn}^{(t)})$
    3. Feedback to Neural Network $f_w(\cdot)$ with $\theta_{inv}^{(t)}$: $w_{inv}^{(t)} = \arg\min_w \mathcal{L}(w; X, Y) + \mathrm{Reg}(w, \theta_{inv}^{(t)})$
    4. Update the clustering kernel $\kappa_c^{(t)}$ with $\theta_{inv}^{(t)}$: $\kappa_c^{(t)} \leftarrow \mathrm{Orthogonal\ Transform}(\kappa_c^{(t-1)}, \theta_{inv}^{(t)})$
**end for**

begin with the invariant prediction step to illustrate our algorithm, and the flow of whole algorithm is shown in figure 1.

## 3.1 $\mathcal{M}_p$: Invariant Gradient Descent with $\mathcal{E}_{learn}$ (Step 1)

For our invariant learning module $\mathcal{M}_p$, we propose Invariant Gradient Descent (IGD) algorithm. Taking the learned environments $\mathcal{E}_{learn}$ as input, our IGD firstly performs invariant learning in Neural Tangent Feature (NTF[10]) space to obtain the invariant direction $\theta_{inv}$, and then guides the whole neural network $f_w(\cdot)$ with $\theta_{inv}$ to learn the invariant model(neural network)'s parameters $w_{inv}$.

**Neural Tangent Feature Space** The NTK theory[10] shows that training the neural network is equivalent to linear regression using non-linear NTFs $\phi(x)$, as in equation 4. For each data point $x \in \mathbb{R}^d$, where $d$ is the feature dimension, the corresponding feature is given by $\phi(x) = \nabla_w f_w(x) \in \mathbb{R}^p$, where $p$ is the number of neural network's parameters. Firstly, we would like to dissect the feature components within $\phi(x)$ by decomposing the invariant and variant components hidden in $\phi(x)$. Therefore, we propose to perform Singular Value Decomposition (SVD) on the NTF matrix:

$$\underbrace{\Phi(X)^T}_{\mathbb{R}^{n \times p}} \approx \underbrace{U}_{\mathbb{R}^{n \times k}} \cdot \underbrace{S}_{\mathbb{R}^{k \times k}} \cdot \underbrace{V^T}_{\mathbb{R}^{k \times p}} \qquad \text{where } p \gg n \geq k \tag{3}$$

Intuitively, in equation 3, each row $V_{j,:}^T$ of $V^T$ represents the $j$-th feature component of $\mathbb{R}^p$ and we take $k$ such different feature components with the top $k$ largest singular values to represent the data, and the rationality of low rank decomposition is guaranteed theoretically[21, 16] and empirically[2]. Since SVD ensures every feature components orthogonal, the neural tangent feature of the $i$-th data point can be decomposed into $\phi(x_i)^T \approx \sum_{j=1}^k U_{i,j} \cdot S_{j,j} \cdot V_{j,:}^T$, where $U_{i,j} \cdot S_{j,j}$ denotes the strength of the $j$-th feature component in the $i$-th data. However, since neural networks have millions of parameters, the high dimension prevents us from learning directly on high dimensional NTFs $\Phi(X)$. Therefore, we rewrite the initial formulation of linear regression into:

$$f_w(X) \approx f_{w_0}(X) + \Phi(X)^T(w - w_0) \approx f_{w_0}(X) + USV^T(w - w_0) \tag{4}$$

$$= f_{w_0}(X) + \Psi(X)\left(V^T(w - w_0)\right) = f_{w_0}(X) + \Psi(X)\theta \tag{5}$$

where we let $\theta = V^T(w - w_0) \in \mathbb{R}^k$ which reflects how the model parameter $w$ utilizes the $k$ feature components. Since $V^T$ is orthogonal, fitting $w - w_0$ with features $\Phi(X)$ is equivalent to fitting $\theta$ using reduced NTFs $\Psi(X)$. In this way, we convert the original high-dimensional regression problem into the low-dimensional one in equation 5, since in wide neural networks, we have $p \gg n \geq k$.

**Invariant Learning with Reduced NTFs $\Psi(X)$** We could perform invariant learning on reduced NTFs $\Psi(X)$ in linear space. In this work, we adopt the invariant regularizer proposed in [11] to learn $\theta = V^T(w - w_0)$ due to its optimality guarantees, and the objective function is:

$$\theta_{inv} = \arg\min_\theta \sum_{e \in \mathcal{E}_{learn}} \mathcal{L}^e(\theta; \Psi, Y) + \alpha \cdot \text{Var}_{\mathcal{E}_{learn}}\left(\nabla_\theta \mathcal{L}^e\right) \tag{6}$$

**Guide Neural Network with invariant direction $\theta_{inv}$** With the learned $\theta_{inv}$, it remains to feed back to the neural network's parameters $w$. Since for neural networks with millions of parameters whose $p \approx 10^8$, it is difficult to directly obtain $w$ as $w = w_0 + V\theta_{inv}$. Therefore, we design a loss function to approximate the projection $(w - w_0 // V\theta_{inv})$. Note that $f_w(X) = f_{w_0}(X) + USV^T(w - w_0) = f_{w_0}(X) + US\theta_{inv}$, we have

$$S^{-1}U^T(f_w(X) - f_{w_0}(X)) = \theta_{inv} \tag{7}$$

Therefore, we can ensure the updated parameters $w$ satisfy that $S^{-1}U^T(f_w(X) - f_{w_0}(X)) \in \mathbb{R}^k$ is parallel to $\theta_{inv}$, which leads to the following loss function:

$$w_{inv} = \arg\min_w \mathcal{L}(w; X, Y) + \lambda \left(1 - \frac{|\langle \theta_{inv}, S^{-1}U^T(f_w(X) - f_{w_0}(X))\rangle|}{\|\theta_{inv}\|\|S^{-1}U^T(f_w(X) - f_{w_0}(X))\|}\right) \tag{8}$$

where $\mathcal{L}(w; X, Y)$ is the empirical prediction loss over training data and the second term is to force the invariance property of the neural network.

## 3.2 Variant Component Decomposition with $\theta_{inv}$ (Step 2)

The core of our KerHRM is the mutual promotion of the heterogeneity exploration module $\mathcal{M}_c$ and the invariant learning module $\mathcal{M}_p$. From our insight, we should leverage the variant components $\Psi_V$

to exploit the latent heterogeneity. Therefore, with the better invariant direction $\theta_{inv}$ learned by $\mathcal{M}_p$ that captures the invariant components in data, it remains to capture better variant components $\Psi_V$ so as to further accelerate the heterogeneity exploration procedure, for which we design a clustering kernel $\kappa_c$ on the reduce NTF space of $\mathbb{R}^k$ with the help of $\theta_{inv}$ learned in section 3.1. Recall the NTF decomposition in equation 3, the initial similarity of two data points $x_i$ and $x_j$ can be decomposed as:

$$\kappa_c^{(0)}(x_i, x_j) = \phi(x_i)^T \phi(x_j) = \langle U_i S, U_j S \rangle \tag{9}$$

With the invariant direction $\theta_{inv}^{(t)}$ learned by $\mathcal{M}_p$ in iteration $t$, we can wipe out the invariant components used by $\theta_{inv}^{(t)}$ via

$$\Psi_V^{(t+1)}(x_i) \leftarrow U_i S - \left\langle U_i S, \theta_{inv}^{(t)} \right\rangle \theta_{inv}^{(t)} / \|\theta_{inv}^{(t)}\|^2 \tag{10}$$

which gives a new heterogeneity-aware kernel that better captures the variant components $\Psi_V^*$ as $\kappa_c^{(t+1)}(x_i, x_j) = \Psi_V^{(t+1)}(x_i)^T \Psi_V^{(t+1)}(x_j)$.

### 3.3 $\mathcal{M}_c$: Heterogeneity exploration with $\kappa_c$ (Step 3)

$\mathcal{M}_c$ takes one heterogeneous dataset as input, and outputs a learned multi-environment partition $\mathcal{E}_{learn}$ for invariant prediction module $\mathcal{M}_p$, and we implement it as a clustering algorithm with kernel regression given the heterogeneity-aware $\kappa_c(x_i, x_j) = \Psi_V(x_i)^T \Psi_V(x_j)$ that captures the variant components in data. Following the analysis above, only the variant components $\Psi_V^*$ should be leveraged to identify the latent heterogeneity, and therefore we use the kernel $\kappa_c$ as well as $\Psi_V(X)$ learned in section 3.2 to capture the different relationship between $\Psi_V^*$ and $Y$, for which we use $P(Y|\Psi_V)$ as the clustering centre. Specifically, we assume the $j$-th cluster centre $P_{\Theta_j}(Y|\Psi_V(X))$ to be a Gaussian around $f(\Theta_j; \Psi_V(X))$ as:

$$h_j(\Psi_V(X), Y) = P_{\Theta_j}(Y|\Psi_V(X)) = (\sqrt{2\pi}\sigma)^{-1} \exp(-(Y - f(\Theta_j; \Psi_V(X)))^2 / 2\sigma^2) \tag{11}$$

For the given $N = \sum_{e \in \text{supp}(\mathcal{E}_{latent})} |D^e|$ data points $D = \{\psi_V(x_i), y_i\}_{i=1}^N$, the empirical distribution can be modeled as $\hat{P}_N = \frac{1}{N} \sum_{i=1}^N \delta_{\psi_V(x_i), y_i}$. Under this setting, we propose one convex clustering algorithm, which aims at finding a mixture distribution in distribution set $\mathcal{Q}$ defined as:

$$\mathcal{Q} = \{Q = \sum_{j \in [K]} q_j h_j(\Psi_V(X), Y), \mathbf{q} \in \Delta_K\} \tag{12}$$

to fit the empirical data best. Therefore, the original objective function and the simplified one are:

$$\min_{Q \in \mathcal{Q}} D_{KL}(\hat{P}_N \| Q) \Leftrightarrow \min_{\Theta, \mathbf{q}} \left\{ \mathcal{L}_c = -\frac{1}{N} \sum_{i \in [N]} \log \left[ \sum_{j \in [K]} q_j h_j(\psi_V(x_i), y_i) \right] \right\} \tag{13}$$

Note that our clustering algorithm differs from others since the cluster centres are learned models parameterized with $\Theta$. As for optimization, we use EM algorithm to optimize the centre parameters $\Theta$ and the mixture weights $q$ iteratively. Specifically, when optimizing the cluster centre model $f(\Theta_j; \cdot)$, we use kernel regression with $\kappa_c(\cdot, \cdot)$ to avoid computing $\Psi_V(X)$ and allow large $k$. For generating the learned environments $\mathcal{E}_{learn}$, we assign $i$-th point to $j$-th cluster with probability $P_{i,j} = q_j h_j(\psi_V(x_i), y_i) / \sum_{l \in [K]} q_l h_l(\psi_V(x_i), y_i)$.

## 4 Theoretical Analysis

In this section, we provide theoretical justifications of the mutual promotion between $\mathcal{M}_c$ and $\mathcal{M}_p$. Since our algorithm does not violate the theoretical analysis in [11] and [10] which proves that better $\mathcal{E}_{learn}$ from $\mathcal{M}_c$ benefits the MIP learned by $\mathcal{M}_p$, to finish the mutual promotion, we only need to justify that better $\theta_{inv}$ from $\mathcal{M}_p$ benefits the learning of $\mathcal{E}_{learn}$ in $\mathcal{M}_c$.

**1. Using $\Psi_V*$ benefits the clustering.** Firstly, we introduce Lemma 4.1 from [15] to show that using $\Psi_V^*$ benefits the clustering in terms of larger between-cluster distance.

**Lemma 4.1.** *For $e_i, e_j \in \text{supp}(\mathcal{E}_{latent})$, assume that $X$ satisfying Assumption 2.1, then under reasonable assumption([15]), we have $D_{KL}(P^{e_i}(Y|X) \| P^{e_j}(Y|X)) \leq D_{KL}(P^{e_i}(Y|\Psi_V^*) \| P^{e_j}(Y|\Psi_V^*))$.*

Then similar to [14], we use the rate-distortion theory to demonstrate why larger $D_{KL}$ between cluster centres benefits our convex clustering as well as the quality of $\mathcal{E}_{learn}$.

**Theorem 4.1.** *(Rate-Distortion) For the proposed convex clustering algorithm, we have:*

$$\min_{Q \in \mathcal{Q}} D_{KL}(\hat{P}_N || Q) = \min_{\Theta} \mathbb{I}(I; J) + (1/2\sigma^2)\mathbb{E}_{I,J}[d(\psi_V(x_i), y_i, \Theta_j)] + Const \qquad (14)$$

*where $r_{ij} = P(j|\psi_V(x_i), y_i)$ is a discrete random variable over the space $\{1, 2, ..., N\} \times \{1, 2, ..., K\}$ which denotes the probability of $i$-th data point belonging to $j$-th cluster, $I, J$ are the marginal distribution of random variable $r_{ij}$ respectively, $d(\psi_V(x_i), y_i, \Theta_j) = (f_{\Theta_j}(\psi_V(x_i)) - y_i)^2$ and $\mathbb{I}(\cdot; \cdot)$ the Shannon mutual information. Note that the optimal $r$ can be obtained by the optimal $\Theta$ and therefore we only minimize the r.h.s with respect to $\Theta$.*

Actually $d$ models the conditional distribution $P(Y|\Psi_V)$. If in the underlying distribution of the empirical data $P(Y|\Psi_V)$ differs a lot between different clusters, the optimizer will put more efforts in optimizing $\mathbb{E}_{I,J}[d(\psi_V(x_i), y_i, \Theta_j)]$ to avoid inducing larger error, resulting in smaller efforts put on optimization of $\mathbb{I}(I; J)$ and a relatively larger $\mathbb{I}(I; J)$. This means data sample points $I$ have a larger mutual information with cluster index $J$, thus the clustering is prone to be more accurate.

**2. Orthogonality Property: Better $\theta_{inv}$ for better $\Psi_V$.** Firstly, we prove the orthogonality property between $\theta_{inv}$(equation 6) and parameters $\Theta$ of clustering centres $f_{\Theta_j}(\cdot)$.

**Theorem 4.2.** *(Orthogonality Property) Denote the data matrix of $j$-th environment $X^j$ and $\Psi_V^j = \Psi_V(X^j)$, then for each $\Theta_j(j \in [K]) = ((\Psi_V^j)^T \Psi_V^j)^{-1}(\Psi_V^j)^T Y^j$, we have $\mathrm{Span}(\Theta) \subseteq \mathrm{Ker}(\theta_{inv})$ and $\mathrm{Span}(\Psi_V^j) \subseteq \mathrm{Ker}(\theta_{inv})$, where $\mathrm{Span}$ denotes the column space and $\mathrm{Ker}$ the null space.*

Theorem 4.2 justifies that the parameter space for clustering model $f_{\Theta}(\cdot)$ as well as the space of learned variant components $\Psi_V$ is orthogonal to the invariant direction $\theta_{inv}$, which indicates that better invariant direction $\theta_{inv}$ regulates better variant components $\Psi_V$ and therefore better heterogeneity. Taking (1) and (2) together, we conclude that better results($\theta_{inv}$) of $\mathcal{M}_p$ promotes the latent heterogeneity exploration in $\mathcal{M}_c$ because of larger between-cluster distance. Finally, we use a linear but general setting for further clarification.

**Example.** *Assume that data points from environments $e \in \mathcal{E}$ are generated as follows:*

$$X = Y(\Psi_S^* + \beta_e \Psi_V^*) + \mathcal{N}(0, \Sigma) \in \mathbb{R}^d \qquad (15)$$

*where $Y = \pm 1$ with equal probability, the coefficient $\beta_e$ varies across environment $e$, $\Psi_S^* \in \mathbb{R}^d$ is the invariant feature and following functional representation lemma [7] $\Psi_V^*$ is the variant feature with $\Psi_V^* \perp \Psi_S^* \in \mathbb{R}^d$ and its relationship with the target $Y$ relies on the environment-specific $\beta_e$.*

**Remark.** *In example 4, when $\mathcal{M}_p$ achieves optimal, we have $\theta_{inv} = \Psi_S^*$, which is the mid vertical hyperplane of the two Gaussian distribution. Then following equation 10, we have $\Psi_V = X - (X^T \theta_{inv})\theta_{inv} = Y\beta_e\Psi_V^*$, which directly shows that in the next iteration, $\mathcal{M}_c$ uses solely variant components $\Psi_V^*$ in $X$ to learn environments $\mathcal{E}_{learn}$ with diverse $P(Y|X) = P(Y|\Psi_V^*)$, which by lemma 4.1 and theorem 4.1 gives the best clustering results.*

# 5 Experiments

In this section, we validate the effectiveness of our method on synthetic data and real-world data.

**Baselines** We compare our proposed KerHRM with the following methods:

- Empirical Risk Minimization(ERM): $\min_\theta \mathbb{E}_{P_{tr}}[\ell(\theta; X, Y)]$

- Distributionally Robust Optimization(DRO [6]): $\min_\theta \sup_{Q \in D_f(Q, P_{tr}) \leq \rho} \mathbb{E}_Q[\ell(\theta; X, Y)]$

- Environment Inference for Invariant Learning(EIIL [5]):

$$\min_{\Phi} \max_u \sum_{e \in \mathcal{E}} \frac{1}{N_e} \sum_i u_i(e)\ell(w \odot \Phi(x_i), y_i) + \lambda \|\nabla_{w|w=1.0} \frac{1}{N_e} \sum_i u_i(e)\ell(w \odot \Phi(x_i), y_i)\|_2 \quad (16)$$

- Heterogeneous Risk Minimization(HRM [15])

- Invariant Risk Minimization(IRM [1]) with environment $\mathcal{E}_{tr}$ labels:

$$\min_{\Phi} \sum_{e \in \mathcal{E}_{tr}} \mathcal{L}^e + \lambda \|\nabla_{w|w=1.0}\mathcal{L}^e(w \odot \Phi)\|^2 \qquad (17)$$

We choose one typical method[6] of DRO as DRO is another main branch of methods for OOD generalization problem of the same setting with us (no environment labels). And HRM and EIIL are another methods for inferring environments for invariant learning without environment labels. We choose IRM as another baseline for its fame in invariant learning, but note that IRM is based on multiple training environments and we provide $\mathcal{E}_{tr}$ labels for it, while the others do not need. Further, for ablation study, we run KerHRM for only one iteration without the feedback loop and denote it as Static KerHRM(KerHRM$^s$). For all experiments, we use a two-layer MLP with 1024 hidden units.

**Evaluation Metrics** To evaluate the prediction performance, for task with only one testing environment, we simply use the prediction accuracy of the testing environment. While for tasks with multiple environments, we introduce Mean_Error defined as Mean_Error $= \frac{1}{|\mathcal{E}_{test}|} \sum_{e \in \mathcal{E}_{test}} \mathcal{L}^e$, Std_Error defined as Std_Error $= \sqrt{\frac{1}{|\mathcal{E}_{test}|-1} \sum_{e \in \mathcal{E}_{test}} (\mathcal{L}^e - \text{Mean\_Error})^2}$, which are mean and standard deviation error across $\mathcal{E}_{test}$. And we use the average mean square error for $\mathcal{L}^e$.

## 5.1 Synthetic Data

**Classification with Spurious Correlation**
Following [19], we induce the spurious correlation between the label $Y \in \{+1, -1\}$ and a spurious attribute $A \in \{+1, -1\}$. Specifically, each environment is characterized by its bias rate $r \in (0, 1]$, where the bias rate $r$ represents that for $100 * r\%$ data, $A = Y$, and for the other $100 * (1-r)\%$ data, $A = -Y$. Intuitively, $r$ measures the strength and direction of the spurious correlation between the label $Y$ and spurious attribute $A$, where larger $|r - 0.5|$ signifies higher spurious correlation between $Y$ and $A$, and $\text{sign}(r - 0.5)$ represents the direction of such spurious correlation, since there is no spurious correlation when $r = 0.5$. We assume $X = H[S, V]^T \in \mathbb{R}^{2d}$, where $S \in \mathbb{R}^d$ is the invariant feature generated from label $Y$ and $V$ the variant feature generated from spurious attribute $A$:

$$S|Y \sim \mathcal{N}(Y\mathbf{1}, \sigma_s^2 I_d), \ V|A \sim \mathcal{N}(A\mathbf{1}, \sigma_v^2 I_d) \tag{18}$$

and $H \in \mathbb{R}^{2d \times 2d}$ is an random orthogonal matrix to scramble the invariant and variant component, which makes it more practical. Typically, we set $\sigma_v^2 \geq \sigma_s^2$ to let the model more prone to use spurious $V$ since $V$ is more informative.

In training, we set $d = 5$ and generate 2000 data points, where $50\%$ points are from environment $e_1$ with $r_1 = 0.9$ and the other from environment $e_2$ with $r_2$. For our method, we set the cluster number $K = 2$. In testing, we generate 1000 data points from environment $e_3$ with $r_3 = 0.1$ to induce distributional shifts from training. In this experiments, we vary the bias rate $r_2$ of environment $e_2$ and the scrambled matrix $H$ which can be an orthogonal or identity matrix (as done in [1]), and results after 10 runs are reported in Table 1.

From the results, we have the following observations and analysis: **ERM** suffers from the distributional shifts between training and testing, which yields the worst performance in testing. **DRO** can only provide slight resistance to distributional shifts, which we think is due to the over-pessimism problem[8]. **EIIL** achieves the best training performance but also performs poorly in testing. **HRM** outperforms the above three baselines, but its testing accuracy is just around the random guess(0.50), which is due to the disturbance of the simple raw feature setting in [15]. **IRM** performs better when the heterogeneity between training environments is large($r_2$ is small), which verifies our analysis in section 2 that the performance of invariant learning methods highly depends on the quality of the given $\mathcal{E}_{tr}$. Compared to all baselines, our **KerHRM** performs the best with respect to highest testing accuracy and lowest (Train_Acc − Test_Acc), showing its superiority to IRM and original HRM.

Further, we also empirically analyze the sensitivity to the choice of cluster number $K$ of our KerHRM. We set $r_2 = 0.80$ and test the performance with $K = \{2, 3, 4, 5\}$ respectively. Results compared with IRM are shown in Table 2. From the results, we can see that the cluster number of our methods does not need to be the ground truth number(ground truth is 2) and our KerHRM is not sensitive to the choice of cluster number $K$. Intuitively, we only need the learned environments to reflect the variance of relationships between $P(Y|\Psi_V^*)$, but do not require the environments to be ground truth. However, we notice that when $K$ is far away from the proper one, the convergence of clustering algorithm is much slower.

**Regression with Selection Bias**
In this setting, we induce the spurious correlation between the label $Y$ and spurious attributes $V$ through selection bias mechanism, which is similar to that in [12]. We assume $X = H[S, V]^T \in$

Table 1: Results in classification simulation experiments of different methods with varying bias rate $r_2$, and scrambled matrix $H$, and each result is averaged over ten times runs.

| $r_2$ | $r_2 = 0.70$ | | $r_2 = 0.75$ | | $r_2 = 0.80$ | |
|---|---|---|---|---|---|---|
| Methods | Train_Acc | Test_Acc | Train_Acc | Test_Acc | Train_Acc | Test_Acc |
| ERM | 0.850 | 0.400 | 0.862 | 0.325 | 0.875 | 0.254 |
| DRO | 0.857 | 0.473 | 0.870 | 0.432 | 0.883 | 0.395 |
| EIIL | **0.927** | 0.523 | **0.925** | 0.470 | **0.946** | 0.463 |
| HRM | 0.836 | 0.543 | 0.832 | 0.519 | 0.852 | 0.488 |
| IRM(with $\mathcal{E}_{tr}$ label) | 0.836 | 0.606 | 0.853 | 0.544 | 0.877 | 0.401 |
| KerHRM$^s$ | 0.764 | 0.671 | 0.782 | 0.632 | 0.663 | 0.619 |
| KerHRM | 0.759 | **0.724** | 0.760 | **0.686** | 0.741 | **0.693** |

Table 2: Ablation study on the cluster number $K$. Each result is averaged over ten times runs.

| | IRM | HRM | KerHRM $(K=2)$ | KerHRM $(K=3)$ | KerHRM $(K=4)$ | KerHRM $(K=5)$ |
|---|---|---|---|---|---|---|
| Train_Acc | 0.877 | 0.852 | 0.741 | 0.758 | 0.756 | 0.753 |
| Test_Acc | 0.401 | 0.488 | 0.693 | 0.687 | 0.698 | 0.668 |

$\mathbb{R}^d$ and $Y = f(S) + \epsilon$, where $f(\cdot)$ is a non-linear function such that $P(Y|S)$ remains invariant across environments while $P(Y|V)$ changes arbitrarily. For simplicity, we select data $(x_i, y_i)$ with probability $P(x_i, y_i)$ according to a certain variable $V_b \in V$:

$$\hat{P}(x_i, y_i) = |r|^{-5*|y_i - \text{sign}(r)*V_b|} \tag{19}$$

where $|r| > 1$. Intuitively, $r$ eventually controls the strengths and direction of the spurious correlation between $V_b$ and $Y$ (i.e. if $r > 0$, a data point whose $V_b$ is close to its $y$ is more probably to be selected.). The larger value of $|r|$ means the stronger spurious correlation between $V_b$ and $Y$, and $r > 0$ means positive correlation and vice versa. Therefore, here we use $r$ to define different environments.

In training, we generate 1000 points from environment $e_1$ with a predefined $r$ and 100 points from $e_2$ with $r = -1.1$. In testing, to simulate distributional shifts, we generate data points for 6 environments with $r \in [-2.9, -2.7, \ldots, -1.9]$. We compare our KerHRM with ERM, DRO, EIIL and IRM. We conduct experiments with different settings on $r$ and the scrambled matrix $H$.

From the results in Table 3, we have the following analysis: **ERM**, **DRO** and **EIIL** performs poor with respect to high average and stability error, which is similar to that in classification experiments(Table 1). The results of **HRM** are quite different in two scenarios, where Scenario 1 corresponds to the simple raw feature setting($H = I$) in [15] but Scenario 2 violates such simple setting with random orthogonal $H$ and greatly harms HRM. Compared to all baselines, our **KerHRM** achieves lowest average error in 5/6 settings, and its superiority is especially obvious in our more general setting(Scenario 2).

**Colored MNIST**
To further validate our method's capacity under general settings, we use the colored MNIST dataset, where data $X$ are high-dimensional non-linear transformation from invariant features(digits $Y$) and variant features(color $C$). Following [1], we build a synthetic binary classification task, where each image is colored either red or green in a way that strongly and spuriously correlates with the class label $Y$. Firstly, a binary label $Y$ is assigned to each images according to its digits: $Y = 0$ for digits $0\sim4$ and $Y = 1$ for digits $5\sim9$. Secondly, we sample the color id $C$ by flipping $Y$ with probability $e$ and therefore forms environments, where $e = 0.1$ for the first training environment, $e = 0.2$ for the second training environments and $e = 0.9$ for the testing environment. Thirdly, we induce noisy labels by randomly flipping the label $Y$ with probability 0.2.

We randomly sample 2500 images for each environments, and the two training environments are mixed without environment label $\mathcal{E}_{tr}$ for ERM, DRO, EIIL, HRM$^s$ and HRM, while for IRM, the $\mathcal{E}_{tr}$ labels are provided. For IRM, we sample 1000 data from the two training environments respectively and select the hyper-parameters which maximize the minimum accuracy of two validation environments. Note that we have no access to the testing environment while training, therefore we cannot resort to testing data to select the best one, which is more reasonable and different from that in [1]. For the others, since we have no access to $\mathcal{E}$ labels, we simply pool the 2000 data points for validation. The results are shown in Table 4, where Perfect Inv. Model represents the oracle

Table 3: Results in selection bias simulation experiments of different methods with varying selection bias $r$, and scrambled matrix $H$, and each result is averaged over ten times runs.

| Scenario 1: Non-Scrambled Setting ($H = I$, varying $r$) | | | | | | |
|---|---|---|---|---|---|---|
| $r$ | $r = 1.5$ | | $r = 1.9$ | | $r = 2.3$ | |
| Methods | Mean_Error | Std_Error | Mean_Error | Std_Error | Mean_Error | Std_Error |
| ERM | 5.056 | 0.223 | 5.442 | 0.204 | 5.503 | 0.234 |
| DRO | 4.571 | 0.205 | 4.908 | 0.180 | 5.081 | 0.209 |
| EIIL | 5.006 | 0.211 | 5.252 | 0.172 | 5.428 | 0.205 |
| HRM | **3.625** | **0.057** | 3.901 | **0.050** | 4.017 | **0.082** |
| IRM(with $\mathcal{E}_{tr}$ label) | 3.873 | 0.176 | 4.536 | 0.172 | 4.509 | 0.194 |
| KerHRM$^s$ | 4.384 | 0.191 | 3.989 | 0.195 | 3.527 | 0.178 |
| KerHRM | 4.112 | 0.182 | **3.659** | 0.186 | **3.409** | 0.174 |
| Scenario 2: Scrambled Setting (random orthogonal $H$, varying $r$) | | | | | | |
| $r$ | $r = 1.5$ | | $r = 1.9$ | | $r = 2.3$ | |
| Methods | Mean_Error | Std_Error | Mean_Error | Std_Error | Mean_Error | Std_Error |
| ERM | 5.059 | 0.229 | 5.285 | 0.207 | 5.478 | 0.211 |
| DRO | 4.494 | 0.212 | 4.717 | 0.175 | 4.978 | 0.207 |
| EIIL | 4.945 | 0.215 | 5.207 | 0.187 | 5.294 | 0.220 |
| HRM | 4.397 | **0.096** | 4.801 | **0.142** | 4.721 | **0.096** |
| IRM(with $\mathcal{E}_{tr}$ label) | 4.269 | 0.218 | 4.477 | 0.174 | 4.392 | 0.178 |
| KerHRM$^s$ | 4.379 | 0.205 | 3.543 | 0.169 | 3.571 | 0.164 |
| KerHRM | **4.122** | 0.195 | **3.375** | 0.163 | **3.473** | 0.160 |

results that can be achieved under this setting. We run each method for 5 times and report the average accuracy, and since the variance of all methods are relatively small, we omit it in the table.

Table 4: Colored MNIST results. The first row indicates whether each method needs the environment label. The Perfect Inv. Model represents the oracle results that can be achieved. The Generalization Gap is defined as (Test Accuracy − Train Accuracy).

| Method | ERM | DRO | EIIL | HRM | IRM | KerHRM$^s$ | KerHRM | Perfect Inv. Model |
|---|---|---|---|---|---|---|---|---|
| Need $\mathcal{E}_{tr}$ Label? | ✗ | ✗ | ✗ | ✗ | ✓ | ✗ | ✗ | - |
| Train Accuracy | **0.845** | 0.644 | 0.777 | 0.835 | 0.766 | 0.802 | 0.654 | 0.800 |
| Test Accuracy | 0.106 | 0.419 | 0.542 | 0.282 | 0.468 | 0.296 | **0.648** | 0.800 |
| Generalization Gap | -0.739 | -0.223 | -0.235 | -0.553 | -0.298 | -0.506 | **-0.006** | - |

From the results, our **KerHRM** generalize the **HRM** to much more complicated data and consistently achieves the best performances. **KerHRM** even outperforms IRM significantly in an unfair setting where we provide perfect environment labels for IRM, which shows the limitation of manually labeled environments. Further, to best show the mutual promotion between $\mathcal{M}_c$ and $\mathcal{M}_p$, we plot the training and testing accuracy as well as the KL-divergence $D_{\mathrm{KL}}$ of $P(Y|C)$ between the learned $\mathcal{E}_{learn}$ over iterations in figure 2. From figure 2, we firstly validate the mutual promotion between $\mathcal{M}_c$ and $\mathcal{M}_p$ since $D_{\mathrm{KL}}$ and testing accuracy escalate synchronously over iterations. Secondly, figure 2 corresponds to our analysis in section 2 that the performance of invariant learning method is highly correlated to the heterogeneity of $\mathcal{E}_{tr}$, which sheds lights to the importance of how to leverage the intrinsic heterogeneity in training data for invariant learning.

## 5.2 Real-world Data

In this experiment, we test our method on a real-world regression dataset (Kaggle) of house sales prices from King County, USA[3], where the target variable is the transaction price of the house and each sample contains 17 predictive variables, such as the built year, number of bedrooms, and square footage of home, etc. Since it is fairly reasonable to assume the relationships between predictive variables and the target vary along the time (for example, the pricing mode may change along the time), there exist distributional shifts in the price-prediction task with respect to the build year of houses. Specifically, the houses in this dataset were built between $1900 \sim 2015$, and we divide the whole dataset into 6 periods, where each contains a time span of two decades. Notice that the later periods have larger distributional shifts. We train all methods on the first period where built_year $\in [1900, 1920)$ and test on the other 5 periods and report the average results over 10 runs in figure 3. For IRM, we further divide the period 1 into two decades for the $\mathcal{E}_{tr}$ provided.

---

[3]https://www.kaggle.com/c/house-prices-advanced-regression-techniques/data

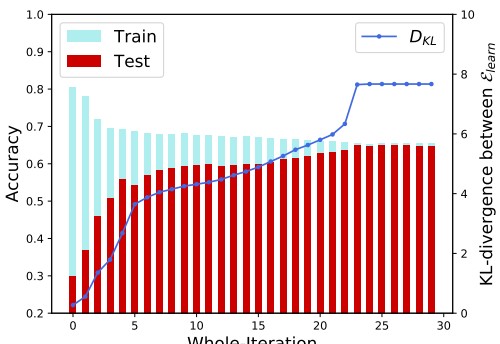
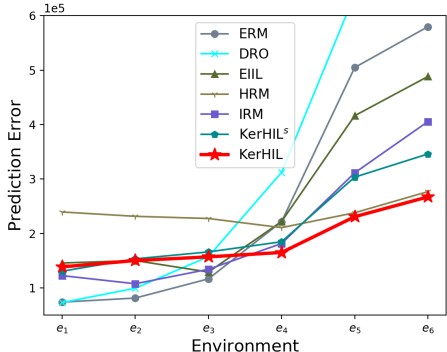

Figure 2: Results for the Colored MNIST task. We plot the training and testing accuracy, as well as the KL-divergence between learned $\mathcal{E}_{tr}$.

Figure 3: Results for the real-word regression task. We train all methods on $e_1$ and test on the others, and report the average results over 10 runs.

**Analysis**  The testing errors of **ERM** and **DRO** increase sharply across environments, indicating the existence of the distributional shifts between environments. **IRM** performs better than ERM and DRO, which shows the usefulness of environment labels for OOD generalization and the possibility of learning invariant predictor from multiple environments. The proposed **KerHRM** outperforms **EIIL** and **HRM**, which validates its superiority of heterogeneity exploration. **KerHRM** even outperforms IRM, which indicates the limitation of manually labeled environments in invariant learning and the necessity of latent heterogeneity exploration.

## 6  Limitations

Although the proposed KerHRM is a competitive method, it has several limitations. Firstly, since in $\mathcal{M}_c$ we take the model parameters as cluster centres, the strict convergence guarantee for our clustering algorithm $\mathcal{M}_c$ is quite hard to analyze. And empirically, we find when the pre-defined cluster number $K$ is far away from the ground-truth, the convergence of $\mathcal{M}_c$ will become quite slow. Further, such restriction also affects the analysis of the mutual promotion between $\mathcal{M}_c$ and $\mathcal{M}_p$, which we can only empirically provide some verification. Besides, although we incorporate Neural Tangent Kernel to deal with data beyond raw feature level, how to deal with more complicated data still remains unsolved. Also, how to reduce the randomness induced by Neural Tangent Kernel and how to incorporate deep learning with the mutual promotion between the two modules needs further investigation, and we left it for future work.

## 7  Conclusion

In this paper, we propose the KerHRM algorithm for the OOD generalization problem, which achieves both the latent heterogeneity exploration and invariant prediction. From our theoretical and empirical analysis, we find that the heterogeneity of environments plays a key role in invariant learning, which is consistent with some recent analysis[17] and opens a new line of research for OOD generalization problem. Our code is available at https://github.com/LJSthu/Kernelized-HRM.

## Acknowledgements

This work was supported National Key R&D Program of China (No. 2018AAA0102004).

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
