## A   Appendix

### A.1   Experimental Details

In this section, we introduce the experimental details as well as additional results. In all experiments, we take $k = \{10, 15, 20, 25\}$ for our KerHIL and select the best one according to the validation results.

**Classification with Spurious Correlation**

For our synthetic data, we set $\sigma_s^2 = 3.0$ and $\sigma_v^2 = 0.3$ to let the model more prone to use spurious $V$ since $V$ is more informative.

**Regression with Selection Bias**

In this setting, the correlations among covariates are perturbed through selection bias mechanism. According to assumption 2.1, we assume $X = H[S, V]^T \in \mathbb{R}^d$ and $S = [S_1, S_2, \ldots, S_{n_s}]^T \in \mathbb{R}^{n_s}$ is independent from $V = [V_1, V_2, \ldots, V_{n_v}] \in \mathbb{R}^{n_v}$ while the covariates in $S$ are dependent with each other. We assume $Y = f(S) + \epsilon$ and $P(Y|S)$ remains invariant across environments while $P(Y|V)$ can arbitrarily change.

Therefore, we generate training data points with the help of auxiliary variables $Z \in \mathbb{R}^{n_s+1}$ as following:

$$Z_1, \ldots, Z_{n_s+1} \overset{iid}{\sim} \mathcal{N}(0, 1.0) \tag{20}$$

$$V_1, \ldots, V_{n_v} \overset{iid}{\sim} \mathcal{N}(0, 1.0) \tag{21}$$

$$S_i = 0.8 * Z_i + 0.2 * Z_{i+1} \quad for \ \ i = 1, \ldots, n_s \tag{22}$$

To induce model misspecification, we generate $Y$ as:

$$Y = f(S) + \epsilon = \theta_s^T S + \beta * S_1 S_2 S_3 + \epsilon \tag{23}$$

where $\theta_s = [\frac{1}{2}, -1, 1, -\frac{1}{2}, 1, -1, \ldots] \in \mathbb{R}^{n_s}$, and $\epsilon \sim \mathcal{N}(0, 0.3)$. For our synthetic data, we set $\beta = 5.0$, $n_s = 5$ and $d = 10$. As we assume that $P(Y|S)$ remains unchanged while $P(Y|V)$ can vary across environments, we design a data selection mechanism to induce this kind of distribution shifts. For simplicity, we select data points according to a certain variable set $V_b \in V$:

$$\hat{P}(x, y) = |r|^{-5*|y-sign(r)*V_b|} \tag{24}$$

$$\mu \sim Uni(0, 1) \tag{25}$$

$$M(r; (x, y)) = \begin{cases} 1, & \mu \le \hat{P} \\ 0, & \text{otherwise} \end{cases} \tag{26}$$

where $|r| > 1$. Given a certain $r$, a data point $(x, y)$ is selected if and only if $M(r; (x, y)) = 1$ (i.e. if $r > 0$, a data point whose $V_b$ is close to its $Y$ is more probably to be selected.) Intuitively, $r$ eventually controls the strengths and direction of the spurious correlation between $V_b$ and $Y$ (i.e. if $r > 0$, a data point whose $V_b$ is close to its $Y$ is more probably to be selected.). The larger value of $|r|$ means the stronger spurious correlation between $V_b$ and $Y$, and $r \ge 0$ means positive correlation and vice versa. Therefore, here we use $r$ to define different environments.

### A.2 Proof of Theorems

### A.2.1 Proof of Theorem 2.1

First, we would like to prove that a random variable satisfying assumption 2.1 is MIP.

**Theorem A.1.** *A representation $\Psi_S^* \in \mathcal{I}$ satisfying assumption 2.1 is the maximal invariant predictor.*

*Proof.* $\rightarrow$: To prove $\Psi_S^* = \arg\min_{Z \in \mathcal{I}} I(Y; Z)$. If $\Psi_S^*$ is not the maximal invariant predictor, assume $\Phi' = \arg\max_{Z \in \mathcal{I}} I(Y; Z)$. Using functional representation lemma, consider $(\Psi_S^*, \Phi')$, there exists random variable $\Phi_{extra}$ such that $\Phi' = \sigma(\Psi_S^*, \Phi_{extra})$ and $\Phi^* \perp \Phi_{extra}$. Then $I(Y; \Phi') = I(Y; \Phi^*, \Phi_{extra}) = I(f(\Psi_S^*); \Psi_S^*, \Phi_{extra}) = I(f(\Psi_S^*); \Psi_S^*)$.

$\leftarrow$: To prove the maximal invariant predictor $\Psi_S^*$ satisfies the sufficiency property in assumption 2.1.

The converse-negative proposition is :

$$Y \ne f(\Psi_S^*) + \epsilon \rightarrow \Psi_S^* \ne \arg\max_{Z \in \mathcal{I}} I(Y; Z) \tag{27}$$

Suppose $Y \ne f(\Psi_S^*) + \epsilon$ and $\Psi_S^* = \arg\max_{Z \in \mathcal{I}} I(Y; Z)$, and suppose $Y = f(\Phi') + \epsilon$ where $\Phi' \ne \Psi_S^*$. Then we have:

$$I(f(\Phi'); \Psi_S^*) \le I(f(\Phi'); \Phi') \tag{28}$$

Therefore, $\Phi' = \arg\max_{Z \in \mathcal{I}} I(Y; Z)$ $\qquad\square$

Then we provide the proof of Theorem 2.1 with Assumption A.1.

**Assumption A.1.** Heterogeneity Assumption.
*For random variable pair $(X, \Phi^*)$ and $\Phi^*$ satisfying Assumption 2.1, using functional representation lemma [7], there exists random variable $\Psi^*$ such that $X = X(\Phi^*, \Psi^*)$, then we assume $P^e(Y|\Psi^*)$ can arbitrary change across environments $e \in \text{supp}(\mathcal{E})$.*

**Theorem A.2.** *Let $g$ be a strictly convex, differentiable function and let $D$ be the corresponding Bregman Loss function. Let $\Psi_S^*$ is the maximal invariant predictor with respect to $I_{\mathcal{E}}$, and put $h^*(X) = \mathbb{E}_Y[Y|\Psi_S^*]$. Under Assumption A.1, we have:*

$$h^* = \arg\min_h \sup_{e \in \text{supp}(\mathcal{E})} \mathbb{E}[D(h(X), Y)|e] \tag{29}$$

*Proof.* Firstly, according to theorem A.1, $\Psi_S^*$ satisfies Assumption 2.1. Consider any function $h$, we would like to prove that for each distribution $P^e (e \in \mathcal{E})$, there exists an environment $e'$ such that:

$$\mathbb{E}[D(h(X), Y)|e'] \geq \mathbb{E}[D(h^*(X), Y)|e] \tag{30}$$

For each $e \in \mathcal{E}$ with density $([\Psi_S, \Psi_V], Y) \mapsto P(\Psi_S, \Psi_V, Y)$, we construct environment $e'$ with density $Q(\Psi_S, \Psi_V, Y)$ that satisfies: (omit the superscript $*$ of $\Psi_S$ and $\Psi_V$ for simplicity)

$$Q(\Psi_S, \Psi_V, Y) = P(\Psi_S, Y)Q(\Psi_V) \tag{31}$$

Note that such environment $e'$ exists because of the heterogeneity property assumed in Assumption A.1. Then we have:

$$\int D(h(\psi_s, \psi_v), y) q(\psi_s, \psi_v, y) d\psi_s d\psi_v dy \tag{32}$$

$$= \int_{\psi_v} \int_{\psi_s, y} D(h(\psi_s, \psi_v), y) p(\psi_s, y) q(\psi_v) d\psi_s dy d\psi_v \tag{33}$$

$$= \int_{\psi_v} \int_{\psi_s, y} D(h(\psi_s, \psi_v), y) p(\psi_s, y) d\psi_s dy q(\psi_v) d\psi_v \tag{34}$$

$$\geq \int_{\psi_v} \int_{\psi_s, y} D(h^*(\psi_s, \psi_v), y) p(\psi_s, y) d\psi_s dy q(\psi_v) d\psi_v \tag{35}$$

$$= \int_{\psi_v} \int_{\psi_s, y} D(h^*(\psi_s), y) p(\psi_s, y) d\psi_s dy q(\psi_v) d\psi_v \tag{36}$$

$$= \int_{\psi_s, y} D(h^*(\psi_s), y p(\psi_s, y) d\psi_s dy \tag{37}$$

$$= \int_{\psi_s, \psi_v, y} D(h^*(\psi_s), y) p(\psi_s, \psi_v, y) d\psi_s d\psi_v dy \tag{38}$$

$$\tag{39}$$

$\square$

### A.2.2 Proof of Lemma 4.1

Firstly, we add the assumption in [18].

**Assumption A.2.** *Assume the pooled training data is made up of heterogeneous data sources: $P_{tr} = \sum_{e \in \text{supp}(\mathcal{E}_{tr})} w_e P^e$. For any $e_i, e_j \in \mathcal{E}_{tr}, e_i \neq e_j$, we assume*

$$I_{i,j}^c(Y; \Phi^*|\Psi^*) \geq \max(I_i(Y; \Phi^*|\Psi^*), I_j(Y; \Phi^*|\Psi^*)) \tag{40}$$

*where $\Phi^*$ is invariant feature and $\Psi^*$ the variant. $I_i$ represents mutual information in $P^{e_i}$ and $I_{i,j}^c$ represents the cross mutual information between $P^{e_i}$ and $P^{e_j}$ takes the form of $I_{i,j}^c(Y; \Phi|\Psi) = H_{i,j}^c[Y|\Psi] - H_{i,j}^c[Y|\Phi, \Psi]$ and $H_{i,j}^c[Y] = -\int p^{e_i}(y) \log p^{e_j}(y) dy$.*

Then the proof for Lemma 4.1 can be found in [18].

### A.2.3 Proof of Theorem 4.1

Firstly, we transform the clustering objective in Equation 12, making it more suitable for further analysis. Proof can be found in [17].

**Theorem A.3.** *Let $\mathcal{Q}'$ be the **set** of distributions of the complete data random variable $(J, \Psi, Y) \in \{1, 2, ..., K\} \times \mathbb{R}^d \times \mathbb{R}$ with elements:*

$$Q'(J = j, \Psi = \psi, Y = y) = q_j h_j(\psi, y), \tag{41}$$

*i.e. $Q'(j, \psi, y)$ is the probability of data point $(\psi, y)$ belonging to the $j$-th cluster. Let $\mathcal{P}'$ be the set of distributions on the same random variable $(J, \Psi, Y)$ which have $\hat{P}_N$ as their marginal on $(\Psi, Y)$. Specifically for any $P' \in \mathcal{P}'$ we have:*

$$P'(j, \psi, y) = \hat{P}_N(\psi, y) P'(j|\psi, y)$$
$$= \begin{cases} \frac{1}{N} r_{ij}, \text{ if } (\phi, y) = (\phi_i, y_i) \\ 0, \text{ otherwise} \end{cases} \tag{42}$$

*where $r_{ij} = P'(j|\psi_i, y_i)$. Then:*

$$\min_{Q \in \mathcal{Q}} D_{KL}(\hat{P}_N || Q) = \min_{P' \in \mathcal{P}', Q' \in \mathcal{Q}'} D_{KL}(P' || Q'). \tag{43}$$

In the new optimization problem in Equation 43, we optimize $P' \in \mathcal{P}'$ and $Q' \in \mathcal{Q}'$. Specifically, in the former we can optimize $r_{ij}$, which is a discrete random variable over the space $\{1, 2, ..., N\} \times \{1, 2, ..., K\}$. Meanwhile, in the latter we can optimize $\{\Theta_j\}_{j=1}^{K}$ and $\{q_j\}_{j=1}^{K}$, which are the cluster centers and cluster weights, respectively.

Substituting the definitions of $P'$ and $Q'$ respective in Equation 42 and Equation 41 to Equation 43, we come the following equation:

$$D_{\text{KL}}(P'||Q') = \frac{1}{N} \sum_{i=1}^{N} \sum_{j=1}^{K} r_{ij}[\log \frac{r_{ij}}{q_j} + \beta d(\psi_i, y_i, m_j)] + \text{Const}, \tag{44}$$

where $\beta = \frac{1}{2\sigma^2}$ is to better illustrate the Rate-Distortion theorem and $d(\psi_i, y_i, \Theta_j) = (f_{\Theta_j}(\psi_i) - y_i)^2$.

It is straightforward to show that for any set of values $r_{ij}$, setting $q_j = \frac{1}{N} \sum_{i=1}^{N} r_{ij}$ minimize the objective, therefore:

$$\begin{aligned} D_{\text{KL}}(P'||Q'^*(P')) = &\frac{1}{N} \sum_{i=1}^{N} \sum_{j=1}^{K} r_{ij}[\log \frac{r_{ij}}{\frac{1}{N} \sum_{i'=1}^{N} r_{i'j}} \\ &+ \beta d(\psi_i, y_i, m_j)] + \text{Const} \\ =&\mathbb{I}(I; J) + \beta \mathbb{E}_{I,J} d(\psi_i, y_i, \Theta_j) + \text{Const}, \end{aligned} \tag{45}$$

where $I, J$ are the marginal distribution of random variable $r_{ij}$ respectively.

The first term is the mutual information between the random variables $I$ (data points) and $J$ (exemplars) under the empirical distribution and the second term is the expected value of the pairwise distances with the same distribution on indices.

Actually $d(\psi_i, y_i, \Theta_j)$ models the conditional distribution $P(Y|\Psi)$. If in the underlying distribution of the empirical data $P(Y|\Psi)$ differs a lot between different clusters, then $d(\psi_i, y_i, \Theta_j)$ will be focused more to be optimized because different clusters are more diverse so the optimizer will put more efforts in optimizing $d(\psi_i, y_i, \Theta_j)$. Resulting in smaller efforts put on optimization of $\mathbb{I}(I; J)$, resulting in a relatively larger $\mathbb{I}(I; J)$. This means data sample points $I$ has a larger mutual information with exemplars $J$, thus the clustering is more accurate.

We can provide another intuition of why larger $\mathbb{I}(I; J)$ means more accurate clustering. For a static dataset to be clustered, setting larger $\beta$ causes distance between points larger, resulting in more clusters which is more accurate. On the other hand, larger $\beta$ signifies the model puts more efforts to optimize $d(\psi_i, y_i, \Theta_j)$ and puts less efforts on the optimization of $\mathbb{I}(I; J)$, resulting in larger $\mathbb{I}(I; J)$.

### A.2.4 Proof of Theorem 4.2

Firstly, since $\Psi_V^{(t+1)}(x_i) \leftarrow U_i S - \left\langle U_i S, \theta_{inv}^{(t)} \right\rangle \theta_{inv}^{(t)}/\|\theta_{inv}^{(t)}\|^2$, we have

$$\left\langle \Psi_V^{(t+1)}, \theta_{inv}^{(t)} \right\rangle = 0 \tag{46}$$

Therefore, we have

$$\mathrm{Span}(\Psi_V^{(t+1)}) \perp \theta_{inv}^{(t)} \tag{47}$$

and

$$\mathrm{Span}(\Psi_V^{(t+1)}) \subseteq \mathrm{Ker}(\theta_{inv}^{(t)}) \tag{48}$$

As for the clustering parameters $\Theta$, since the kernel regression is equivalent to linear regression using mapping function $\Psi_V$, we can directly derive the analytical solution of $\Theta_j$ as:

$$\Theta_j(j \in [K]) = ((\Psi_V^j)^T \Psi_V^j)^{-1} (\Psi_V^j)^T Y^j \tag{49}$$

where $\Psi_V^j$ denotes the data matrix of environment $j$ and $Y^j$ the corresponding label matrix. Then since

$$((\Psi_V^j)^T \Psi_V^j)^{-1} (\Psi_V^j)^T Y^j = (\Psi_V^j)^T ((\Psi_V^j)(\Psi_V^j)^T)^{-1} Y^j \tag{50}$$

we have

$$\Theta_j^T \theta_{inv} = \left[ (\Psi_V^j)^T ((\Psi_V^j)(\Psi_V^j)^T)^{-1} Y^j \right]^T \theta_{inv} \tag{51}$$

$$= 0 \tag{52}$$

which gives the conclusion.

### A.3 Limitations and Future Work

This work focus on the integration of latent heterogeneity exploitation and invariant learning on representation level. To fulfill the mutual promotion between environment inference and invariant learning, we give up deep learning for representation learning, since the representation space in deep learning is hard to theoretically analyzed, which makes it quite hard to maintain the property we need. As an alternative, we leverage Neural Tangent Kernel(NTK) and convert data into Neural Tangent Feature(NTF) space, for NTK theory[13] builds the equivalency between MLP and kernel regression.

However, we have to admit that using NTF space for representation space is not as powerful as the representation space produced by recent deep learning methods. But we would like to emphasize the difficulty in incorporating deep learning, since we cannot directly use the learned representation for heterogeneity exploitation, because during the invariant representation learning process, deep models will gradually extract the latent invariant components $\Psi_S^*$ in data and discard those variant components $\Psi_V^*$. We have to resort to variant components $\Psi_V^*$ rather than invariant ones $\Psi_S^*$ to explore the heterogeneity, but variant components are discarded during the training of deep models. Therefore, incorporating deep learning while maintaining mutual promotion is quite hard and we leave it for future work.

### A.4 Related Work

There are mainly two branches of methods for OOD generalization problem, namely Distributionally Robust Optimization(DRO) methods[6, 8, 22, 24] and Invariant Learning methods[1, 3, 5, 14, 18].

To ensure the OOD generalization performances, DRO methods[6, 8, 22, 24] aim to optimize the worst-performance over a distribution set, which is usually characterized by $f$-divergence or Wasserstein distance. However, in real scenarios, it is often necessary for the distributional set to be large to contain the potential testing distributions, which results in the over-pessimism problem because of the large distribution set[10, 11].

Realizing the difficulty of solving OOD generalization problem without prior knowledge or structural assumptions, invariant learning methods assume the existence of causally invariant relationships and propose to explore them through multiple environments. However, the effectiveness of such methods relies heavily on the quality of training environments. Further, modern big data are frequently assembled by merging data from multiple sources without explicit source labels, which results in latent heterogeneity in pooled data and renders these invariant learning methods inapplicable.

Recently, there are methods[5, 18] aiming at relaxing the need for multiple environments for invariant learning. [5] directly infers the environments according to a given biased model first and then performs invariant learning. But the two stages cannot be jointly optimized and the quality of inferred environments depends heavily on the pre-provided biased model. Further, for complicated data, using invariant representation for environment inference is harmful, since the environment-specific features are gradually discarded, causing the extinction of latent heterogeneity and rendering data from different latent environments undistinguishable. [18] designs a mechanism where two interactive modules for environment inference and invariant learning respectively can promote each other. However, it can only deal with scenarios where invariant and variant features are decomposed on raw feature level, and will break down when the decomposition can only be performed in representation space(e.g., image data).