# OpenReview forum: "Integrated Latent Heterogeneity and Invariance Learning in Kernel Space"
_NeurIPS.cc/2021/Conference — NeurIPS 2021 Poster_

### Official Review · Reviewer_Rd4w · 2021-07-11

**Rating:** 6
**Confidence:** 4

**Summary:**

The paper proposes KerHIL, a new invariant learning method for out-of-distribution (OOD) generalization. KerHIL both explores the latent heterogeneity and learns the invariance in kernel space by iterating between two modules M_c and M_p, respectively.

**Limitations And Societal Impact:**

None.

**Main Review:**

Despite some typos and references errors (in checklist), the paper is well-written. The authors explained and demonstrated the method clearly, and gave theoretical proofs for the theorems. Experiments are also done in a good manner. However, there are still some points I am curious about.

First of all, the authors stressed the difficulty brought by multi-source training datasets. However, this issue is only discussed empirically using training dataset with only two sources. It would be better if the authors could extend the training sources to three or more.

Secondly, it would be more intuitive if the authors provide explanations or even guesses about how predicting training environment labels aid the training process. And how accurate the prediction need to be in order to fully help the training.

For the experiments conducted, experiment details are well listed. However, the parameter settings for different baselines and the proposed KerHIL are not demonstrated. For example, the specific value for the regularization term in Equation (8) is not included. What’s more, it would be good if the authors add ablation studies so as to explore more about the properties of KerHIL. This can help to identify what is the bottleneck (which module) for KerHIL and may provide intuitions about how to extend the method to more complicated datasets and deeper neural networks.

Overall, I think the paper gives a great direction for future research while the current efforts seem somehow preliminary, and need further improvement in both theoretical and experimental parts.

**Time Spent Reviewing:**

6

---

> ### Author Response · Authors · 2021-08-10
> **Response to Reviewer Rd4w**
>
> Thank you for your insightful and positive feedback! We address your comments below and will incorporate all feedback in the final version.
>
> **R4.1. Extend the training sources**: Firstly, our method does not require the number of training source to be 2 and can easily extend to multiple sources. Further, we add extra results with multiple training sources on our first simulation data(Table 1) as following, for time reason, we only compare KerHIL and ERM.
>
> | | KerHIL Train | KerHIL Test| ERM Train| ERM Test|
> |----|----|----|----|----|
> |3 Sources($r=0.9,0.8,0.7$)|0.751|**0.743**| 0.849| 0.408|
> |4 Sources($r=0.95,0.90,0.85,0.80$)|0.771|**0.732**|0.888|0.205|
> which shows that our methods can handle multiple training sources, and we will add more results in final version.
>
> **R4.2. How predicting training environment labels aid the training process**: Firstly, our clustering module aims to cluster different relationships between $\Psi_V(X)$ and $Y$, which we formulate as the conditional distribution $P(Y|\Psi_V(X))$. Then with the learned environments, $E_{learn}$, our backend module can leverage the heterogeneity among $E_{learn}$ to exclude the variant component(e.g. style, color) and better capture the invariant direction(e.g. digits, shape).
>
> **R4.3. How accurate does the prediction needs to be**:
>   * Firstly, we argue that the accuracy may not be the perfect criterion to quantify the quality of clustering results, because the ground-truth or manually labeled environment labels are not guaranteed to be optimal. For example, in the standard setting of Colored MNIST where $e_1=0.1$ and $e_2=0.2$, the two training environments both contain spurious correlation between $Y$ and color in the same direction. However, the optimal or best environments should be like $e_1=0$ and $e_2=1$, that is, the directions of the spurious correlation between $Y$ and color are opposite, which makes models easily distinguish such spurious correlation and abandon the color/style for prediction.
>   * Secondly, we think the proper criterion of the clustering results is to quantify the heterogeneity or discrepancy between $E_{learn}$(e.g. The distance between $P_1(Y|C)$ and $P_2(Y|C)$ in Color MNIST). Because good environments should produce smaller       |$\mathcal{I}$($\mathcal{E}$)|, and the more heterogeneous or diverse of $P(Y|\Psi_V)$ among environments, the smaller the invariance set |$\mathcal{I}$($\mathcal{E}$)|.
>   * In Figure 2, we plot the curve of $D_{KL}(P_1(Y|C)\|P_2(Y|C))$ to show how the quality of clustering results changes along with iterations. From the results, we can see that the quality is growing better and the OOD performance becomes better synchronously with the quality of environments. We can also see that, if the quality of environments is better than random split(start point of the curve), the OOD performance is better, and the promotion of OOD performance is closely related to the quality of environments.
>
>
>
> **R4.4. Ablation studies**:
>   * Firstly, in our paper, the baseline KerHIL$^s$ serves as an ablation study. One key contribution of our KerHIL is the mutual promotion between $\mathcal{M}_c$ and $\mathcal{M}_p$, and KerHIL$^s$ serves to empirically verify such mutual promotion. KerHIL$^s$ is a static version of KerHIL which only runs for one iteration and does not have the feedback loop. By comparing the performance between KerHIL$^s$ and KerHIL, we can see that there does exist mutual promotion between our $\mathcal{M}_c$ and $\mathcal{M}_p$ to help achieve better results.
>   * Secondly, we also add one ablation study to better analyze our $\mathcal{M}_c$ and $\mathcal{M}_p$. We introduce a new baseline called IRM$^s$, where we run our clustering algorithm once to generate environments, and then use IRM to make invariant prediction. For time reasons, we test this new baseline on our first simulation data(Table 1):
>
>   |Test Acc|$r_2=0.70$|$r_2=0.75$|$r_2=0.80$|
>   |----|----|----|----|
>   |EIIL(Cluster Once+IRM)|0.523|0.470|0.463|
>   |IRM$^s$(Our $\mathcal{M}_c$ Once + IRM)|0.596|0.587|0.545|
>   |KerHIL$^s$(Our $\mathcal{M}_c$ Once + Our $\mathcal{M}_p$)|0.677|0.588|0.617|
>   |KerHIL(Our $\mathcal{M}_c$+$\mathcal{M}_p$+mutual promotion)|**0.727**|**0.677**|**0.669**|
>
> * From the results, we can see that our clustering method is effective to explore the heterogeneity inside data(compare IRM$^s$ with EIIL), and our proposed invariant gradient descent $\mathcal{M}_p$ is effective than IRM(compare KerHIL$^s$ with IRM$^s$), and the mutual promotion does exist(compare KerHIL with KerHIL$^s$).
>
>
> **R4.5. Hyper-parameters**: For IRM and EIIL, we select the $\lambda\in${1e-1, 3e-1, 5e-1, 1e0, 5e0, 1e1, 3e1, 6e1, 1e2, 3e2, 5e2}(larger in MNIST and smaller in others). For DRO, we select the $\eta\in${1e-2,5e-2,1e-1,3e-1,5e-1} and $k\in${2,3,4}. For KerHIL, we select the $\lambda\in$  {5e-2, 1e-1, 5e-1, 1e0, 5e0, 1e1, 2e1} and cluster number $K\in${2,3,4,5}, and please refer to **R1.1** and **R2.1** for detail. We will add all experimental details in final version.

---

### Official Review · Reviewer_iPzu · 2021-07-14

**Rating:** 7
**Confidence:** 1

**Summary:**

The authors present an algorithm for invariance learning under distributional shifts which greatly improves OOD generalization. This is an extremely important problem, and the authors give both good analysis and empirical results.

**Limitations And Societal Impact:**

I have big no comments here, but I hope the authors will include more limitations in the final version of the paper.

**Main Review:**

Section 2 is a really present read. Unfortunately, section 1 is almost unreadable to readers without this prior knowledge. I would suggest the authors restructure, such that the "related work" section comes after you have defined what you mean by invariant learning etc.

At line 165 you lose me. What is big theta_j ? Does delta denote Dirac's delta?

How is it determined how many clusters (or sources) there are?

Can the authors expand why KerHIL^s is of importance?

Do you think the results on Colored MNIST are affected by you using an MLP for your predictor? Would the color have as much influence if a CNN was used?

Overall, I think the results are really good and you test out all the relevant hypotheses with a positive outcome. Very nice!

I did not verify mathematical proofs.

**Time Spent Reviewing:**

2.5

---

> ### Author Response · Authors · 2021-08-10
> **Response to Reviewer iPzu**
>
> Thank you for your insightful and positive feedback! We address your comments below and will incorporate all feedback in the final version.
>
> **R3.1. Notations in line 165**: $\Theta_j$ is the model parameters for the $j$-th clustering model $f(\Theta_j;\Psi_V(X))$, and $\delta$ is the Dirac's delta.
>
> **R3.2. How is it determined how many clusters (or sources) there are**: Actually how to control the cluster number is an important problem in clustering algorithms and we think there is no definitive answer to this question[1], especially for our proposed clustering algorithm which takes models as cluster centres. But we can provide some intuitions empirically that if the cluster number is far away from the proper one, the convergence of the clustering would be quite slow, which may empirically do some help. As for our experiments, we select $K\in${2,3,4,5} according to the R.H.S of equation (14), which quantifies how well the cluster models fit the empirical data distribution. Further, for future work, we will try to incorporate convex clustering methods[2], which can automatically choose the cluster number.
>
> **R3.3. Why KerHIL$^s$ is of importance**: KerHIL$^s$ serves as an ablation study. One key contribution of our KerHIL is the mutual promotion between $\mathcal{M}_c$ and $\mathcal{M}_p$, and KerHIL$^s$ serves as an ablation study to empirically verify such promotion. KerHIL$^s$ is a static version of KerHIL which only runs for one iteration and does not have the feedback loop. By comparing the performance between KerHIL$^s$ and KerHIL, we can see that there does exist mutual promotion between our $\mathcal{M}_c$ and $\mathcal{M}_p$ to help achieve better results.
>
> **R3.4. Whether CNN is biased by color**: CNN models will be affected by color too. We add results for CNN models(LeNet-like architecture), and the results over 10 runs are as following:
>
> | | Train | Test|
> |----|----|----|
> |MLP|0.845|0.106|
> |CNN|0.851|0.103|
>
> [1] Kodinariya, T. M., & Makwana, P. R. (2013). Review on determining number of Cluster in K-Means Clustering.
>
> [2] Chi, E. C., & Lange, K. (2015). Splitting methods for convex clustering.

---

> > ### Comment · Reviewer_iPzu · 2021-08-22
> > **Response**
> >
> > I am happy with the authors' response and maintain my score, especially if the authors include the discussion of how K affects convergence.

---

> > > ### Author Response · Authors · 2021-08-23
> > > **Response to Reviewer iPzu**
> > >
> > > Thanks for your time and feedback! We will add some empirical results and discuss the convergence in the final version.

---

### Official Review · Reviewer_w5jL · 2021-07-16

**Rating:** 7
**Confidence:** 5

**Summary:**

This paper proposes a novel optimization framework for Out-Of-Distribution (OOD) generalization problems. OOD generalization problem is one of the most important problems in today's machine learning,  and it is highly non-trivial to investigate it at the optimization framework level.
The proposed framework focuses on the integration of latent heterogeneity discovery and invariant learning on representation level, by introducing neural tangent kernel (NTK). Theoretical analysis is also conducted to verify the mutual promotion of these two modules. Experiments show that the proposed method can significantly outperform the commonly used and latest optimization algorithms.

**Limitations And Societal Impact:**

Yes

**Main Review:**

This paper is well-written overall with a clear positioning with regard to the related research literature. I am convinced that this problem is relevant and critical to the ICML community. Although the overall logic follows the Heterogeneous Risk Minimization proposed in [15], this paper makes an important step forward by weakening the assumption that invariant and variant features can be decomposed on raw feature level. I like in particular how the proposed framework is theoretically justified. The heterogeneity exploration part is analyzed with a rate-distortion perspective, and the orthogonality property of kernels is guaranteed.  Experiment detail is clear in the paper and appendix on dataset details, baseline methods, and how to choose the hyperparameters for the proposed method. The results are satisfactory overall, and some of them are quite impressive.

The weakness of the proposed method would be the requirement of the hyperparameter to control the cluster number. I would like to see how the number of clusters affects the performance. Also, the authors are encouraged to provide some failure case analysis. For example, whether the mutual promotion between these two modules is sensitive to initialization.

I believe this paper has merits that outweigh flaws and I would like to vote for acceptance.

**Time Spent Reviewing:**

7

---

> ### Author Response · Authors · 2021-08-10
> **Response to Reviewer w5jL**
>
> Thank you for your insightful and positive feedback! We address your comments below and will incorporate all feedback in the final version.
>
> **R2.1. Requirement of the hyperparameter to control the cluster number**: Actually how to control the cluster number is an important problem in clustering algorithms and we think there is no definitive answer to this question[1], especially for our newly proposed clustering algorithm which takes models as cluster centres. But empirically we can provide some intuitions that if the cluster number is far away from the proper one, the convergence of the clustering would be quite slow, which may empirically guide the selection. As for our experiments, we select $K\in${2,3,4,5} according to the R.H.S of equation (14), which quantifies how well the cluster models fit the empirical data distribution. Further, for future work, we will try to incorporate convex clustering methods[2], which automatically choose the cluster number.
>
> **R2.2. How the number of clusters affects the performance**: Our method is not sensitive to the choice of cluster number $K$. We add some extra experimental results to better analyze the influence of cluster number $K$. For the first simulation data(Table 1), we set $r_2=0.8$ and test the performance with $K=${2,3,4,5} respectively. Results averaged for 10 runs are as following.
>
>   |     | IRM  | HRM| $K=2$| $K=3$|$K=4$|$K=5$|
>   |  ----  | ----  | ----| ----| ---- | ----| ---- |
>   | Train Acc  | 0.877 | 0.852| 0.762| 0.758|0.756|0.753|
>   | Test Acc  | 0.401 | 0.488| **0.669**|**0.687**|**0.698**|**0.668**|
>
>   From the results, we can see that the cluster number of our methods does not need to be the ground truth number(ground truth is 2) and our KerHIL is not sensitive to the choice of cluster number $K$. Intuitively, we only need the learned environments to reflect the variance of relationships between $P(Y|\Psi_V)$, but do not require the environments to be ground truth. However, we notice that when $K$ is far away from the proper one, the convergence of the clustering algorithm is much slower.
>
> **R2.3. Some failure case analysis**: We think there exist failure cases for weak latent heterogeneity. If data are little heterogeneous but strongly biased(for example, almost all images with $Y = 0$ are red and with $Y = 1$ are green), our clustering module is possible to be misled to use invariant $\Psi_S$(e.g., shape) naturally(which may due to the fact that between-class distance is larger than between-environment distance), and KerHIL might continuously produce wrong environments. However, as shown in our empirical results, such extreme cases are rare, and even it occurs, other invariant learning methods will also fail to generalize well. As for more commonly occured weak heterogeneity cases where data are not strongly biased, KerHIL will perform similar to ERM.
>
> [1] Kodinariya, T. M., & Makwana, P. R. (2013). Review on determining number of Cluster in K-Means Clustering.
>
> [2] Chi, E. C., & Lange, K. (2015). Splitting methods for convex clustering.

---

### Official Review · Reviewer_Rqt2 · 2021-07-16

**Rating:** 8
**Confidence:** 5

**Summary:**

This paper studies the out-of-distribution (OOD) generalization problem. It proposes a Kernel heterogeneity & Invariance Learning method, which jointly learns both the latent heterogeneity exploration and invariant learning in kernel space. This paper provides strong theoretical guarantees for the proposed method. We know the proposed method makes clear improvements over related baselines in OOD scenarios from the empirical results.

**Limitations And Societal Impact:**

Yes

**Main Review:**

The proposed studied in this paper is an important and promising direction. This paper first tries to jointly learn latent heterogeneity environments and invariance learning modules on representation level. It is important for applying invariance learning on unstructured data (e.g., image).
By introducing Neural Tangent Kernel (NTK) into the proposed algorithm, the authors convert non-linear neural networks into linear regression in NTK. Thus, mutual promotion will be well guaranteed. Moreover, a novel invariant gradient descent method is proposed to learn an invariant classifier. Meanwhile, the environment-specific features are captured by the orthogonal heterogeneity-aware kernel to accelerate the heterogeneity exploration. Thus, the proposed method is both novel and theoretically sound.
The paper conducts extensive experiments on synthetic and real-world data to validate the effectiveness of the proposed method. The proposed outperforms baselines with a large margin on OoD datasets.
Questions:
1.	How to select the hyperparameter lambda in Eq.(8)?
2.	Could the proposed framework integrate the deep model (e.g., deep CNN) as the backbone to deal with a more complex image dataset?


**Time Spent Reviewing:**

12

---

> ### Author Response · Authors · 2021-08-10
> **Response to Reviewer Rqt2**
>
> Thank you for your insightful and positive feedback! We address your comments below and will incorporate all feedback in the final version.
>
> **R1.1 Selection for $\lambda$ in Equation (8)**: Equation (8) is the last procedure in our framework, which propagates the learned $\theta_{inv}$ to the neural network parameters $w_{inv}$. In experiments, we select the $\lambda\in$  {5e-2, 1e-1, 5e-1, 1e0, 5e0, 1e1, 2e1} according to the average validation accuracy and the gap of training accuracy among learned environments.  As Gulrajani et al.[1] pointed out, it lacks a standard criterion for choosing hyper-parameters in OOD generalization problems, we make our own rules that we select the hyper-parameter with maximal average validation accuracy subjected to the maximal gap of training accuracy among $\mathcal{E}_{learn}$ is smaller than 5%.
>
> **R1.2 Integrate the deep model as the backbone**: Our method can be extended to deep models. Firstly, the proposed KerHIL uses Neural Tangent Kernel(NTK) to linearize the MLP, and some recent progress[2,3,4,5] in NTK shows that deep models(like CNN) can also be linearized, which indicates the proposed KerHIL can be extended to deep models. Further, we think another potential way for incorporating deep models is to leverage recent pre-trained models, which we leave for future work[6].
>
> [1]Gulrajani, I., & Lopez-Paz, D. (2020). In search of lost domain generalization.
>
> [2]Arora, S., Du, S. S., Hu, W., Li, Z., Salakhutdinov, R. R., and Wang, R. On exact computation with an infinitely wide neural net.
>
> [3]Yang, G. (2020). Tensor programs ii: Neural tangent kernel for any architecture.
>
> [4]Alemohammad, S., Wang, Z., Balestriero, R., & Baraniuk, R. (2020). The recurrent neural tangent kernel.
>
> [5]Novak, R., Xiao, L., Hron, J., Lee, J., Alemi, A. A., Sohl-Dickstein, J., & Schoenholz, S. S. (2019). Neural tangents: Fast and easy infinite neural networks in python.
>
> [6]Achille, A., Golatkar, A., Ravichandran, A., Polito, M., & Soatto, S. (2021). Lqf: Linear quadratic fine-tuning.

---

### Decision · Program_Chairs · 2021-09-27

**Decision:**

Accept (Poster)

**Comment:**

This paper studies the OOD generalization. The paper proposes kernel heterogeneity and the invariance learning method, which enable to learn the latent heterogeneity and invariant learning in kernel space.  The proposed method and theory provide novel and interesting directions for the OOD problem.  The empirical results are also good.   This work deserves to be presented in NeurIPS.